# Coexistence of Genetic Diseases Is a New Clinical Challenge: Three Unrelated Cases of Dual Diagnosis

**DOI:** 10.3390/genes14020484

**Published:** 2023-02-14

**Authors:** Anna Paola Capra, Maria Angela La Rosa, Sara Briguori, Rosa Civa, Chiara Passarelli, Emanuele Agolini, Antonio Novelli, Silvana Briuglia

**Affiliations:** 1Department of Chemical, Biological, Pharmaceutical and Environmental Sciences, University of Messina, Viale Ferdinando Stagno D’Alcontres 31, 98166 Messina, Italy; 2Genetics and Pharmacogenetics Unit, “Gaetano Martino” University Hospital, Via Consolare Valeria 1, 98125 Messina, Italy; 3Translational Cytogenomics Research Unit, Bambino Gesù Children’s Hospital, IRCCS, 00165 Rome, Italy; 4Department of Biomedical and Dental Sciences and Morphofunctional Imaging, University of Messina, Via Consolare Valeria 1, 98125 Messina, Italy

**Keywords:** genetic diseases, phenotype, clinical competence, diagnosis, rare diseases

## Abstract

Technological advancements in molecular genetics and cytogenetics have led to the diagnostic definition of complex or atypical clinical pictures. In this paper, a genetic analysis identifies multimorbidities, one due to either a copy number variant or a chromosome aneuploidy, and a second due to biallelic sequence variants in a gene associated with an autosomal recessive disorder. We diagnosed the simultaneous presence of these conditions, which co-occurred by chance, in three unrelated patients: a 10q11.22q11.23 microduplication and a homozygous variant, c.3470A>G (p.Tyr1157Cys), in the *WDR19* gene associated with autosomal recessive ciliopathy; down syndrome and two variants, c.850G>A; p.(Gly284Arg) and c.5374G>T; p.(Glu1792*), in the *LAMA2* gene associated with merosin-deficient congenital muscular dystrophy type 1A (MDC1A); and a de novo 16p11.2 microdeletion syndrome and homozygous variant, c.2828G>A (p.Arg943Gln), in the *ABCA4* gene associated with Stargardt disease 1 (STGD1). The possibility of being affected by two relatively common or rare inherited genetic conditions would be suspected when signs and symptoms are incoherent with the primary diagnosis. All this could have important implications for improving genetic counseling, determining the correct prognosis, and, consequently, organizing the best long-term follow-up.

## 1. Introduction

Medical genetics encompasses many different areas, and it is becoming increasingly relevant to the diagnosis of many common and rare genetic diseases. The particularity of some genetic diseases underlines the value of adequate clinical experience; moreover, it emphasizes how a multidisciplinary approach is essential in managing and taking care of patients [1,2]. Each patient undergoes a diagnostic evaluation tailored to their clinical signs and symptoms, generating a differential diagnosis and using appropriate genetic tests to provide an accurate diagnosis [3]. Complex chromosomal rearrangements are the most important cause of multiple congenital anomaly/intellectual disability (MCA/ID) syndromes. In the suspicion of genomic syndromes, investigations can provide the standard associated karyotype with the use of a chromosomal microarray analysis (CMA), for the detection of genetic imbalances. In complex cases, in-depth analyses using new technologies, such as next-generation sequencing (NGS), have made it possible to identify gene mutations and new disease-causing genes.

The possibility of being affected by two, relatively common or rare, inherited genetic conditions should be considered when findings are incoherent with the primary diagnosis. In these patients, a complete diagnosis improves their clinical management and, of course, fully explains their atypical phenotype. According to the definition of comorbidities indicating the coexistence of two or more disorders in the same individual, we describe the coexistence of two different genetic conditions in three unrelated patients. There are three main ways in which different diseases may be found in the same individual: by chance, selection bias or one or more type of causal association [4]. The analysis performed proved that the co-occurrence of two genetic disorders was by chance and without a causative relationship.

We diagnosed the coexistence of a maternal 10q11.22q11.23 microduplication and autosomal recessive ciliopathy caused by a homozygous variant in the *WDR19* gene; a case of Down syndrome with a congenital muscular disease associated with two variants in the *LAMA2* gene; and a de novo 16p11.2 microdeletion and Stargardt disease caused by a homozygous variant in the *ABCA4* gene.

The rarity of these disease associations underlines the value of adequate genetic counseling; moreover, it emphasizes how a multidisciplinary approach is essential in managing and treating patients to determine the most correct prognosis and, consequently, organize the best long-term care.

## 2. Materials and Methods

We evaluated three patients with clinical, phenotypic, and blood sampling for genetic testing. The clinical evaluation included a physical examination, familial context, other blood sampling, and instrumental tests collected during the genetic counseling. Written consent was obtained from all participants and parents. We withdrew a blood sample for genetic tests and performed a karyotype examination. A cell culture of peripheral lymphocytes was produced, and G-banded karyotyping at a resolution of 400–500 banding was performed, both according to standard protocols. Twenty metaphases per individual were analyzed. Karyograms were prepared using Ikaros, a computer-assisted karyotyping system (MetaSystems Hard & Software GmbH, Altlußheim, Germany). The karyotype was described according to the guidelines in the International System of Human Cytogenetic Nomenclature (ISCA 2020). Further investigation with a CMA was performed. Genomic DNA from patients was obtained from peripheral blood using a QIAamp DNA Blood Mini Kit (Qiagen, Hilden, Germany). The quantity of the DNA samples was determined using a Qubit 2.0 fluorometer with a Qubit dsDNA BR Assay Kit. A CMA was performed on the genomic DNA using an Agilent platform (8 × 60K or 4 × 180K oligonucleotide array; Agilent Technologies) and analyzed using the Agilent Cytogenomics software (v. 5.0.2.5). Labeling, hybridization, and data processing were carried out according to the manufacturer’s recommendations. Benign copy number variations (CNVs), as reported in the Database for Genomic Variants (http://dgv.tcag.ca/dgv/app/home accessed on 16 November 2020), were removed from the results. Parental DNA samples were examined with a CMA for all patients. The second step of the analysis used NGS, and was supported with clinical consideration and rethinking in order to provide a more accurate diagnosis and effective genetic counseling. Enrichment and parallel sequencing were performed on genomic DNA extracted from circulating leukocytes of the affected subjects and their unaffected parents. A library preparation was carried out using a custom SeqCap EZ Choice Enrichment Kit, according to the manufacturer’s protocols (Roche NimbleGen, Inc.; Madison, WI, USA), and sequenced on a NextSeq550 (Illumina) platform. The BaseSpace pipeline (Illumina, https://basespace.illumina.com/ accessed on 1 November 2020) and the TGex software (LifeMap Sciences, http://tgex.genecards.org/ accessed on 16 November 2020) were used for variant calling and annotating variants, respectively. Sequencing data were aligned to the hg19 human reference genome. The variants were analyzed in silico using the Sorting Intolerant from Tolerant (SIFT), Polymorphism Phenotyping v2 (PolyPhen-2), Mutation Taster, and Combined Annotation Dependent Depletion (CADD) tools for the prediction of deleterious nonsynonymous single-nucleotide variants for human diseases. Based on the guidelines of the American College of Medical Genetics and Genomics, a minimum depth coverage of 30× is considered suitable for analysis. Variants were examined for coverage and Q-score (with a minimum threshold of 30) and visualized with the Integrative Genomics Viewer (IGV). Mutations identified as pathogenic were confirmed with Sanger sequencing, following a standard protocol (BigDye Terminator v3.1 Cycle Sequencing Kit, Applied Biosystems by Life Technologies).

## 3. Results

We described three Caucasian subjects whose diagnoses were first determined based on the main signs and symptoms, such as cognitive disabilities, dysmorphological phenotypes, and major malformations. We performed the appropriate genetic analyses, with cytogenetic or molecular approaches set up according to the primary clinical suspicion.

### 3.1. Case 1

The patient was an 8-year-old child, born from third-degree consanguineous parents. He suffered from chronic renal failure, polycystic kidney disease, and polycystic liver disease. Brain magnetic resonance imaging (MRI) was normal, except for trigonocephaly. The patient was born at term, with delivery complicated by facial and cranial deformities, a prolonged labor, and neonatal cyanosis. His psychomotor development was delayed and the onset of walking was reported at approximately 18 months of age. He was determined to have an intellectual disability. The electrocardiogram was normal, while the echocardiogram showed the dysplasia of the aortic valve associated with the aneurysmal dilatation of the ascending aorta. The ophthalmological examination showed myopia and marked vascular tortuosity with arterial venous crossings at the fundus oculi.

The clinical examination showed trigonocephaly, microcephaly (<3rd centile), a short stature (high 114 cm), overweight (24.4 kg), and skin syndactyly of the II and III fingers and toes.

The CMA revealed a 5.3 Mb heterozygous duplication of 10q11.22q11.23 (GRCh37/hg19 chr10:46264302–51595050), which was maternally inherited. The mother was healthy and asymptomatic. The duplicated region contained different RefSeq genes: *RBP3*, *GDF2*, *ERCC6*, *CHAT*, *SLC18A3*, and *MSMB*. We performed a review of the literature and curated databases, ClinVar and DECIPHER. In ClinVar, the reported CNV was interpreted as pathogenic with incomplete penetrance. As the 10q11 duplication could not explain the renal and liver polycystic diseases, we performed clinical exome sequencing, which showed a novel homozygous variant, NM_025132.3: c.3470A>G (p.Tyr1157Cys), in the WDR19 gene on chromosome 4p14. The described homozygous variant, p.Tyr1157Cys, in the WDR19 gene was inherited from his heterozygous parents. *WDR19* is a ciliary gene that encodes the intraflagellar transport 144 (IFT144) protein. The missense variant, c.3470A>G, was reported on in the dbSNP (rs1735198326) and ClinVar databases, but not in the HGVD nor the Genome Aggregation Database (gnomAD). This variant was not previously reported on in the literature in individuals with *WDR19*-related conditions.

### 3.2. Case 2

This patient was the first child of nonconsanguineous parents. The mother was 40 years old, and an amniocentesis, performed for advanced maternal age, resulted in a diagnosis of free 21 chromosome trisomy, or Down syndrome 47,XY+21. The pregnancy was completed, and he was born with Down syndrome, complicated by a type A ventricular atrium canal, which was surgically treated, severe neonatal hypotonia, gastroesophageal reflux, hypovalid sucking, severe growth deficiency, left cryptorchidism, required enteral feeding with a nasogastric tube for gavage, and, subsequently, a pump. Brain ultrasound, electroencephalogram (EEG), and electromyography results were normal, enabling treatment to continue. For the persistence of severe generalized hypotonia, a severe psychomotor delay, and the absence of head control after approximately 6 months of therapy, the dosage of CPK was increased by 15–20 times. At the age of 17 months, a lateral muscle biopsy documented congenital muscular dystrophy, not due to dystrophinopathy, and which is currently undergoing diagnostic definition. A targeted NGS gene panel showed pathogenetic variants in the *LAMA2* gene (NM_000426.3) on chromosome 6q22, inherited by the healthy heterozygous parents. The patient was a compound heterozygote with c.850G>A; p.(Gly284Arg) and c.5374G>T; p.(Glu1792*). The first is a missense variant in exon 6 of the gene that affects the V protein domain; the second creates a premature translational stop signal causing the loss of normal protein function, either through protein truncation or nonsense-mediated mRNA decay [5,6]. The reported pathogenic variants were previously described in affected homozygous patients.

### 3.3. Case 3

The third patient was a 27-year-old man who was the son of third-degree consanguineous parents. At the age of 17 years, he was evaluated due to a clinical picture characterized by generalized obesity, a growth impairment, delay in psychomotor development, epilepsy, myopia, a dysmorphic phenotype, relative lengthening of the ears, gynecomastia, hypogonadism, an alteration in the curve of the spine, a tapered appearance of the fingers, and alterations in some toes. An echocardiogram showed slight insufficiencies of the mitral, tricuspid, and aortic valves. At the age of 21 years, a CMA showed a de novo heterozygous 16p11.2 microdeletion of 205 Kb (GRCh37/hg19 chr16:28837450-29042118). The deleted region contains different RefSeq genes: *ATXN2L*, *TUFM*, *SH2B1*, *ATP2A1*, *RABEP2*, *CD19*, *NFATC2IP*, *SPNS1,* and *LAT*.

At the age of 24 years, due to a deterioration in his vision, a new ophthalmological examination diagnosed bilateral macular dystrophy and diffuse opacities of the lens. An optical coherence tomography (OCT), electroretinogram (ERG), and visual evoked potential (VEP) diagnosed Stargardt maculopathy. A genetic test provided a definitive diagnosis, identifying a homozygous variant, NM_000350: c.2828G>A (p.Arg943Gln), in the *ABCA4* gene on chromosome 1p22.1, which he inherited from his healthy heterozygous parents. The variant Arg943Gln (rs1801581) is reported on the gnomAD database with an allelic frequency of 0.03012, its functional significance being controversial. Indeed, it has been detected both in healthy subjects and in mild forms of age-related macular degeneration [7].

## 4. Discussion

Due to rapid advances in genomic technologies, genetic analyses have become essential in clinical practice. A correct clinical evaluation allows for the use of an appropriate genetic test in order to try to identify the cause of a condition, even in complex phenotypes or in the presence of more than one inherited disease. Moreover, accurate genetic counseling is essential in managing and treating patients. In this paper, we focused on the coexistence of two unrelated genetic conditions, whose diagnoses were conducted, initially, with cytogenetic and cytogenomic tests, and, later, by carrying out a NGS analysis. We reported on three cases of genetic coexistence between chromosomal aneuploidy, or CNVs, and rare pathogenetic variants in well-known disease genes. For instance, the diagnosis of Down syndrome was clinically suspected and confirmed in one patient with standard karyotyping. In the other cases, after normal karyotyping results, a CMA was performed, reporting two different CNVs. The CNVs found were associated with the pathologic phenotype (see Appendix A). One was confirmed to be of maternal origin; the other one was de novo. However, the diagnosis was insufficient in completely clarifying the clinical phenotype. The appearance or the persistence of unexpected and unforeseen symptoms in the natural history of the already diagnosed chromosomal diseases led us to a more in-depth study, carried out with additional molecular tests using NGS. A sequencing data analysis revealed three Mendelian diseases, caused by recessive genes *WDR19*, *LAMA2,* and *ABCA4*. This strategy allowed for the detection of all the chromosome abnormalities and disease-causing variants present in an unusual and complex phenotypic presentation of coexisting genetic conditions.

Case one presented a maternal 10q11.22q11.23 microduplication and a ciliopathy caused by a homozygous missense variant in the *WDR19* gene.

Comparisons of the phenotypes with an overlapping genomic location, maternally inherited in 46,XY patients (cases #272785 and #337896) showed an intellectual disability, short stature, atrial septal defect, hypoplasia of the corpus callosum, and neonatal hypotonia. Case #383145, described on DECIPHER as 46,XX, with a 5.51 Mb microduplication at 10:46292022-51804949, showed an abnormality of the nervous system, intellectual disability, low-set ears, macrodontia, microcephaly, micrognathia, and a short stature. Variants in the *WDR19* gene have been associated with heterogeneous recessive diseases, described in patients with cranioectodermal dysplasia 4, asphyxiating thoracic dystrophy 5, isolated or combined nephronophthisis 13, and retinitis pigmentosa (Senior–Loken syndrome 8) [8]. Cilia participate in signaling pathways that transmit information within and between cells, and are important for the development and function of many types of cells and tissues, including cells in the kidneys, the liver, and the light-sensitive tissue at the back of the eye (the retina). During intraflagellar transport, cells use molecules called IFT particles, which contain IFT–A and IFT–B complexes, to carry materials to and from the tips of cilia. This is essential for the assembly and maintenance of cilia. The protein encoded by *WDR19* is part of IFT complex A [9,10]. Recently, it was shown that the IFT144 protein, coded by the *WDR19* gene, together with IFT122, coded by *WDR140*, constituted the interface of the IFT–A complex and the IFT–B complex, and that this interaction plays a crucial role in ciliary protein trafficking. Thus, different missense mutations in the *WDR19* gene can affect protein–protein interactions, and this can lead to different clinical manifestations associated with ciliary dysfunction [9]. The effect of this missense change on the protein structure and function was shown to be damaging with the use of several bioinformatics tools, including PolyPhen-2, Align-GVGD, SIFT, i-Mutant, and PROVEAN, with a scaled CADD score of 32. Multiple-sequence alignment performed with the ConSurf Job and Uniprot Align tools, showed that Tyr1157 was evolutionarily conserved in all the vertebrates we examined, suggesting its functional importance in the architecture of proteins. Moreover, it is important to consider the physicochemical difference between tyrosine and cysteine. In gnomAD, the same residue was reported with a different substitution, aspartic acid instead of cysteine, with an allele frequency of 0.000004014, and also, in this case, the variant effect predictor suggested a probably damaging/deleterious effect. To date, ClinVar has reported on 417 variants in the *WDR19* gene, of which approximately 13% are pathogenic/likely pathogenetic, 30% are benign/likely benign, and the majority, nearly 57%, were reported as having uncertain/conflicting clinical significance. All of these variants suggest that Tyr1157Cys is likely to be disruptive, but this was not confirmed in published functional studies, so the available evidence is currently insufficient to determine the clinical significance of this variant. In this study, we described, for the first time, a patient with c.3470A>G (p.Tyr1157Cys) homozygosity, characterized by minimal skeletal features, a delayed psychomotor development, and the development of juvenile isolated nephronophthisis in the presence of cystic kidney disease, without overt ocular pathology. This clinical picture also overlaps with the large microduplication reported in 10q11.22q11.23, mainly associated with a short stature, microcephaly, and the dysplasia of the aortic valve.

Case two was affected by Down syndrome and a rare congenital muscular disease related to a homozygous variant of the *LAMA2* gene. This gene encodes the α2 chain subunit of laminin 2 (merosin) and laminin 4 (s-merosin), so the second diagnosis of this patient was early-onset *LAMA2*-related muscular dystrophy (OMIM#156225). The signs and symptoms appear at birth or within the first few months of life, including contractures of the large joints, profound hypotonia, poor spontaneous movement, and severe muscle weakness [11]. This case showed the coexistence of a common genetic condition, such as Down syndrome, correlated with advanced maternal age, and a very rare autosomal recessive muscular dystrophy inherited from unrelated healthy carrier parents. There was a coincidence in the two independent events, being that one is commonly associated with an increased risk in older pregnant women, and one as being very rare in the general population, with a prevalence of 1–9 per 1,000,000 children [12].

Case three was a boy who had 16p11.2 microdeletion syndrome and the subsequent onset of Stargardt disease. 16p11.2 microdeletion syndrome (OMIM #611913) is a rare genetic disorder with a population prevalence of approximately 1/5000. The associated phenotypic spectrum includes autism, mild intellectual disability/developmental delay, and/or possibly other primary psychiatric disorders, with nonspecific major or minor dysmorphisms [13]. Recently, ophthalmic manifestations, a highly penetrant form of obesity, recurrent infections, and extensive phenotypic variability, have also been reported in a literature review in association with this microdeletion syndrome [14]. At the age of 24, the patient was diagnosed with Stargardt maculopathy (OMIM#248200, prevalence: 1/8,000–10,000). The *ABCA4* gene encodes for the retinal ATP-binding cassette (ABC) transporter protein, which transports all transretinal in rod or cone outer segment cells during the phototransduction cascade using the energy released through nucleotide hydrolysis. Functional assessments demonstrated that both ATPase and GTPase activities of the mutated protein Arg943Gln were reduced compared to wild-type ones, even with a low impact [7,15]. In this case, two rare inherited conditions coexisted with different temporal onsets: a congenital disease and a late-onset disease.

Different possible mechanisms are reported in a complex clinical picture when two or more conditions coexist in one patient, including direct causation, associated risk factors, heterogeneity, and independence. The co-occurrence of two independently inherited genetic diseases is a peculiar condition of comorbidity, most often defined in relation to a specific index condition [4,16]. When multiple diseases are present in one individual, it can be referred to as a multimorbidity [16]. The three unrelated patients described were examples of multimorbidity in genetic diseases for the simultaneous presence of two hereditary conditions with different demonstrated genetic causes. In particular, they received two different genetic diagnoses, one related to a copy number variant or chromosomal aneuploidy and another due to biallelic sequence variants in a gene associated with an autosomal recessive disorder. The morbidity burden was worsened by the combined impact of different genetic diseases in each individual, taking into account their severity. The clinical complexity of these patients was greatly increased, leading to worse health outcomes, more complex clinical management, and increased healthcare costs [17].

## 5. Conclusions

In the future, considering the power of biotechnology approaches in medicine, the detection of genetic multimorbidity could become even more frequent, as well as in a prenatal setting or neonatal–perinatal screening.

In clinical practice, considering how many different types of genetic tests are available, it is fundamental to take into consideration several factors for selecting the appropriate test, keeping in mind what conditions are clinically suspected and the genetic variations typically associated with those conditions. If a diagnosis is unclear, or the primary pathogenetic condition does not fully explain the phenotype of a patient, a more focused test may be performed in combination to reach a complete diagnosis. The right diagnostic approach becomes very important in improving genetic counseling, determining the most correct prognosis, and, consequently, permitting for the organization of the best clinical management. Moreover, carrier testing for at-risk relatives and prenatal testing for pregnancies at increased risk is now possible if pathogenic variants in the family have previously been identified. All this could also have important implications for clinical research and public health.

## Data Availability

Data of this study are available to the corresponding author’s address.

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
