# Peer review of "Coexistence of Genetic Diseases Is a New Clinical Challenge: Three Unrelated Cases of Dual Diagnosis"

_genes, 2023, doi:10.3390/genes14020484_

Round 1
Reviewer 1 Report
In this paper, authors report three interesting cases of double genetic diagnosis. As stated in the introduction, I think it is of high interest for the reader to see how multidisciplinary discussion about atypical cases can lead to improved diagnosis accuracy and thus to better health care for the patient.
However, I have two main points of concern.
First, text structure should be corrected.
For copy number variants, boundaries and encompassed genes should be in results rather than on discussion. The same for point variants nomenclatures, allele frequencies, algorithmic pathogenicity predictions, and parental status (inherited or de novo).
In the methods, first sentence (line 64) seems useless and inappropriate in this section. This section also mixes past and present tenses. Authors should also avoid carriage return when they do not change subject. For instance, lines 69-75 should be one paragraph to make reading easier (this problem also occurs in the “discussion” section).
Second, and more importantly, pathogenicity of genetic alterations reported is sometimes unclear or ambiguous. Authors should for instance clarify by using “pathogenic variant” or “class 5 variant” if they do have enough elements to conclude on pathogenicity. If missense variants or copy number variant remain of “unknown significance” (class 3) it is not possible to formally state that clinical symptoms are “due” to the genetic alteration (for instance in the introduction). These cases are still of interest, but authors should be more cautious in their conclusions. To say it in another way: reader should know which genetic findings can be used for genetic counselling in these patients.
In case 1:
It is still unclear to me if inherited 10q11 duplication is pathogenic or not. I understand that few patients are reported in DECIPHER but are the symptoms specific enough, and the duplication rare enough, to conclude ? Does the mother have any minor clinical sign?
I understand that WDR19 missense variant is of unknown significance. It should be clearly stated in the “results” section, with all the arguments in favor of pathogenicity. Maybe there is enough arguments to consider it as “class 4” (probably pathogenic)?
Important point (lines 221-229): note that algorithmic pathogenic predictions (SIFT, PolyPhen, Mutation taster, CADD) are partially redundant between each other as well as with statements that Tyr1157 is evolutionarily conserved, that Tyr and Cys are physicochemicaly different, and that “variant effect predictor suggests a probably damaging/deleterious effect”.
Line 125 : I would propose « As the 10q11 duplication could not explain renal and liver polycystic disease, we performed clinical exome 125 sequencing » rather than « For the presence of renal and liver polycystic disease, we performed clinical exome 125 sequencing ». But it seems in the discussion that other symptoms of this patient could be due du WDR19 (paragraphs line 183 and line 197)?
If doable, a table in supplementary data with all symptoms of the patient and for each symptom either if it is “certainly” or “possibly” due to each genetic alteration would help the reader a lot. A similar table could also be done for the two other cases, especially case 3.
Case 2:
Pathogenicity of both genetic alterations is clearer. LAMA2 variant soulhd be mention as a “pathogenic variant”. For instance, line 251: « a missense pathogenic variant, reported at homozygous state in affected patients »
Case 3:
I understand that ABCA4 variant is ambiguous. Therfore world “mutation” (line 279) should be avoided. Number of homozygous in gnomAD (182) should be precised (line 280). As for Case 1, I think which symptom is certainly or possibly due to each genetic alteration should be clarified.
Other minor comments:
1) “We focused on the association of other genetic factors using Next-generation se- 86 quencing (NGS) technology.” (line 86) should be reformulated or removed.
2) Abstract line 20: “one due to either a copy number variant or a chromosome aneuploidy” instead of “genomic disorder such as a copy number variant or chromosome aneuploidy”
3) Avoid redundancy between paragraph line 183 and line 197.
4) Line 129: why do amniocentesis have been performed? Only because of maternal age or because of any prenatal signs?
Author Response
We appreciate the time and effort that you have dedicated to providing your valuable feedback on our manuscript. We have been able to incorporate changes to reflect most of the suggestions provided by the reviewers.
We have highlighted the changes within the manuscript.
The extended commentary on the CNVs and variants found in the described patients were included in the results, as suggested.
We have, accordingly, deleted first sentence (line 64) and revised the tenses in the section. Moreover, carriage returns were edited in all the manuscript to facilitate reading.
About the second point, as suggested, we presented the CNVs and variants better and more extensively in the results section, and then resumed and commented their clinical interpretation and impact in the discussion.
In case 1:
Thank you for pointing this out. We agree with this comment. Therefore, we have described the mother’s phenotype (line 130) and reported all the clinical symptoms in the supplementary table.
The WDR19 missense variant remains of uncertain clinical significance, it is very infrequent, as reported in gnomAD database. In the clinical case described, it is inherited in homozygous state from healthy heterozygous parents, and peculiar clinical symptoms seems to be associated to this genetic alteration in the class of ciliopathies.
As proposed, we modified the sentence “As the 10q11 duplication could not explain renal and liver polycystic disease, we performed clinical exome sequencing”.
In case 2:
Thank you for this suggestion. We emphasized the evident pathogenicity of the mutations found in compound heterozygosity in the LAMA2 gene. The reported variants were previously described in affected homozygous patients.
In case 3:
We agree with this and we have incorporated the description of controversial ABCA4 variant in the result section. We better defining its clinical impact in the discussion and reporting the associated symptoms in the suggested supplementary table to improve understanding of the clinical picture in association with genetic anomalies.
Other minor comments:
1) As suggested, the sentence was reformulated.
2) Agree. We have, accordingly, revised the sentence in abstract.
3) Thank you for this suggestion. We incorporate changes in the text to avoid redundancy between paragraph line 183 and line 197.
4) Thank you for this comment. The amniocentesis was performed for advanced maternal age, so we adjust the sentence to clarify this aspect (line 145).
Reviewer 2 Report
Well-documented description of three patients with dual cytogenomic findings, each providing a novel DNA variant for the literature.
I would put the aCGH and full variant nomenclature of each DNA change in the abstract beside the cited gene, so search engines will pick them up (patients with combined variants may not be registered in searches on the particular gene-region. More importantly, the discussion should mention the possibility of widespread neonatal screening when multiple DNA aberrations will be commonplace and apply their recommendation of distilling findings for the appropriate phenotype-genotype correlations to that future.
Author Response
We appreciate the time and effort that you have dedicated to providing your valuable feedback on our manuscript. We have been able to incorporate changes to reflect most of the suggestions provided by the reviewers.
We have highlighted the changes within the manuscript.
We agree with your suggestions and we have incorporated CNV and genetic variants in the abstract, moreover, accordingly, we modified the discussion to mention the possibility of widespread neonatal screening (line 296).